# The Hard Reality of Biogas Production through the Anaerobic Digestion of Algae Grown in Dairy Farm Effluents

Marianne Hull-Cantillo, Mark Lay *, Graeme Glasgow and Peter Kovalsky

School of Engineering, University of Waikato, Private Bag 3105, Hamilton 3240, New Zealand;
mh351@students.waikato.ac.nz (M.H.-C.); graeme.glasgow@waikato.ac.nz (G.G.);
peter.kovalsky@waikato.ac.nz (P.K.)
* Correspondence: mark.lay@waikato.ac.nz

**Abstract:** Much emphasis has been given to algal biomass growth in dairy farm wastewater. Most of the systems examined require productive land to be converted and/or freshwater use to dilute high concentrations of nutrients found in dairy effluent. A rotating algal biofilm (RABR) provides the capacity to grow algae without sacrificing productive land or freshwater. In theory, this system would overcome some of the economic and environmental challenges that other systems have. A combination of theoretical information, nutrient uptake formulas, and economic formulas were used to calculate the potential of biogas production from algae grown in an RABR with dairy effluents. The average nutrient uptake was 0.8 mgN/m$^2$ per day and 0.1 mgP/m$^2$ per day. The maximum methane production from the anaerobic digestion of algae was 112 m$^3$/RABR·year. The minimum and maximum economic scenarios resulted in gross profits of NZD −2101 and −1922. After evaluating this system for the first time in the New Zealand dairy farming context, it was found that biogas production from an RABR is not a feasible option for New Zealand dairy farmers.

**Keywords:** anaerobic digestion; dairy; effluent; biogas; New Zealand; rotating algal biofilm reactor; algae

## 1. Introduction

Growing algae in dairy effluent is a well-studied topic [1–3]. Different configurations, such as open and closed reactors, have been developed. Attached methods of growing algae include filamentous algae nutrient scrubbers (FANSs) and algal turf scrubbers (ATSs). A study performed on filamentous algae growth on anaerobically digested food waste resulted in biomass productivities of 17.4 to 27.8 g/m$^2$·day and nutrient uptakes of 0.3 to 0.6 gN/m$^2$·day and 0.06 to 0.08 gP/m$^2$·day at different harvesting times. An economic comparison of dairy effluent that had been previously digested and a non-digested effluent used to grow algae in an ATS was studied [4]. The system-treated effluent from 1000 cows used 11 hectares of land. The profit and expense per cow for a 1000-cow farm using this system were as follows: USD 500 profit/cow and USD 454 expense/cow for a system coupled with anaerobic digestion and USD 631 expense/cow for a system without any pretreatment.

High-rate algal ponds have been used in municipal wastewater with biomass productivity values of 95 g/m$^3$·day [5]. The removal efficiencies of this system are 11 gN/m$^3$·day and 1.6 gP/m$^3$·day [5]. In New Zealand, high-rate algal ponds (HRAPs) [6] have been studied for dairy effluent treatment. Nevertheless, they are not commonly found on dairy farms. A 15-year study performed in New Zealand on wastewater treatment with HRAP showed that the cost-effective and sustainable harvesting of biomass is still a barrier to the implementation and adoption of this technology [7].

In addition to the costs of some of the production systems, there are two other factors that could limit its application to New Zealand: the first one is the need to dilute the dairy

effluent with freshwater in order to use algal turf scrubbers [2]; the second one is the amount of productive land that would be required in order to set up this system. An option that would not require freshwater addition or extending the land used for effluent treatment would be a rotating algal biofilm reactor (RABR) in an existing effluent pond. Between 2011 and 2012, different researchers applied the concept of rotating biological contractors (RBCs) to algae. This resulted in the development of RABRs, an attached growth system, which comprise a rotating cylinder and allow for algae to grow on the cylinder surface while it rotates between wastewater and air [8]. A study by Christenson et al. [8] showed that the biomass productivity in RABR can be between 12.6 and 23.6 $g/m^2 \cdot$day higher than that of suspended cultures in raceways grown in municipal water. Some studies have focused on the application of this system for lipid production [9], biodiesel and methane production from algae grown on municipal wastewater [10], and growth on dairy wastewater [11].

Different researchers have focused on the production of biogas from algae [12–21]. Some have used dairy effluent for algal growth and the further digestion of the produced biomass [22]. Others have focused on the co-digestion of algal biomass and bovine effluent [23–25]. One of the most studied taxonomic groups is green algae. Biogas production from the anaerobic digestion of green algae was found to be in the range of 178–540 mL $CH_4$/gVS [22]. Different studies have shown results for the co-digestion of the algae and dairy effluent. A combination of 25% fresh *U. lactuca* and 75% dairy slurry resulted in a biogas production of 178 L $CH_4$/kgVS [23]. Meanwhile another study showed production values of 292 mL $CH_4$/gVS for *Chroococcus* sp. and cow manure on a 1:1 VS basis. The biogas production range for dairy effluent on its own has been in the range of 265–343 L $CH_4$/kgVS for dairy manure, solid fraction, and liquid fraction [26].

Currently, no published work has focused on algal growth on an RABR in dairy farm wastewater for the sole purpose of producing biogas. In this study, the biogas production from algae grown in an RABR in the liquid portion of dairy wastewater after proceeding through solid separation is examined. In addition, the application of RABR to New Zealand dairy effluent has never been studied. This study evaluates the economic potential of using this technology. In addition, the bioremediation capacity, which is the amount of nitrogen and phosphorus that algae can uptake, has been evaluated.

## 2. Materials and Methods

### 2.1. Rotating Algal Biofilm Reactor

A system resembling a rotating algal biofilm reactor (RABR) and a floating deck were designed and positioned directly in a typical New Zealand dairy effluent pond (Figure 1). The economics, energy requirements, and bioremediation capacity were theoretically evaluated. For the application of an RABR, dairy effluent information from a previous study [27] was used to determine the nutrient loading, solid composition, and volumes for a 410-cow farm, which is close to the average size farm in New Zealand [28]. A sensitivity analysis was performed to account for the different values found in the literature for algal growth, biogas production, and nutrient absorption.

Aluminum was used as the material for the design of the rotating wheel. This was mainly due to its lower weight when compared to steel and other materials. The floating board for the wheel was also designed with aluminum and eight 57 L plastic drums. A DC brushless motor with a maximum output torque of 15 Nm and a power rating of 73 W was chosen to rotate the wheel continuously. In addition, a Panasonic $^{TM}$ spur gearbox (MZ9G20B, Radio Spares, New Zealand) was used to provide the required rotational speed of 1.2 RPM. After the system design was theoretically developed, an economic assessment was carried out.

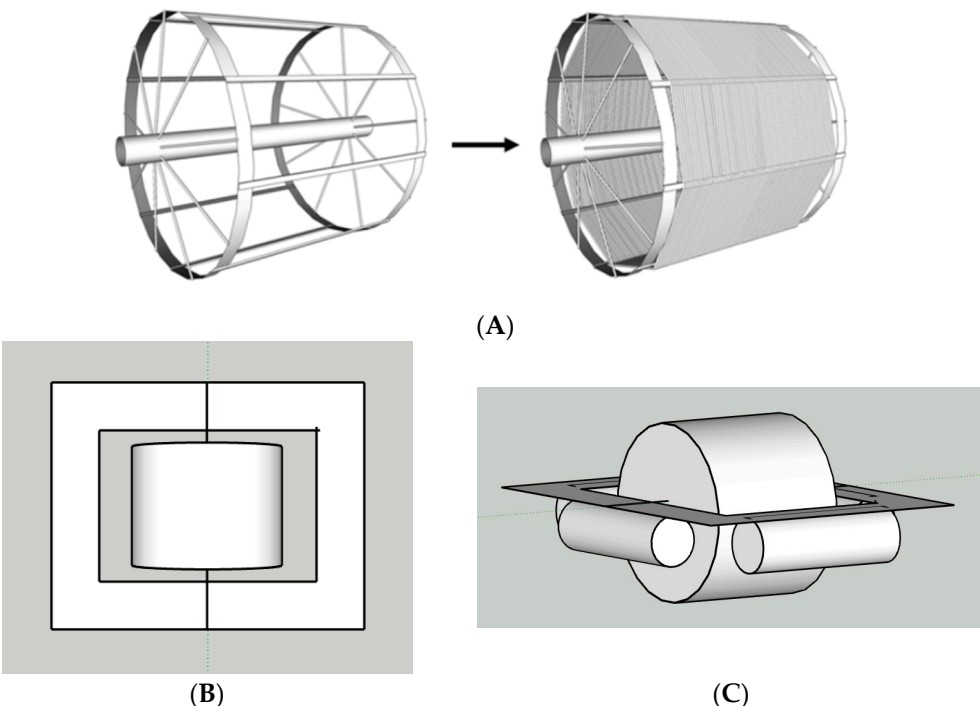

**Figure 1.** The floating RABR design. (**A**) The wheel designed without and with cotton string. Reproduced with permission from Logan Christenson. Rotating algal biofilm reactor and spool harvester for wastewater treatment with biofuel by-products; published by biotechnology and bioengineering, 2012. (**B**) Top view of the floating platform with the wheel connected to it. (**C**) Side view of the floating platform.

Different materials have been studied as mediums for algae attachment in an RABR. In a study comparing algal production on nylon, polypropylene, cotton, jute, and acrylic ropes, as well as high-thread and low-thread cotton sheets, it was found that cotton rope was the best substrate for algal production [8]. The growth in this material was between 20 and 30 g/m$^2$·day. Therefore, this was the chosen material used for the design of the floating RABR.

### 2.2. Algae Selection

A study conducted in the United States on an existing RABR in a wastewater treatment plant contained *Pseudanabaena* sp., *Oscillatoria* sp., and *Chroococcus* sp. as the main algal species [11]. In this study, the daily biomass production was between 5.15 and 8.69 g/m$^2$·day. These values depended on the amount of total organic carbon (TOC) present in the water. For a TOC value of 1200 mg/L, the growth was 8.69 g/m$^2$·day. The case study farm used in this study had a TOC of 1380 mg/L. The values from this previous study were extrapolated to predict what the growth of algae with a TOC content of 1380 mg/L would be using the following formula:

$$\text{Biomass productivity} = (0.0039 \times \text{TOC}) + 4.025 \tag{1}$$

where biomass is given in units of g/m$^2$·day, and TOC is in units of mg/L. The R$^2$ value of this correlation is 0.998. Using this correlation, we found the biomass productivity to be 9.41 g/m$^2$·day for a TOC content of 1380 mg/L. Another factor that affected the productivity of algae and was studied by Fica and Sims [11] is temperature. In their study, the following formula was developed to calculate the biomass production based on temperature:

$$K \ prediction = 5.152 \times \theta^{T_{water} - 280} \tag{2}$$

where *K* prediction is the biomass production in g/m²·day, $\theta$ is the coefficient related to the TOC content, and the temperature of water ($T_{water}$) is in degrees, Kelvin. The temperature range studied was between 7 and 27 °C. In a study performed by Park et al. [29], it was found that the lowest dairy effluent pond temperature was 8 °C in winter, and the highest was 26 °C in the summer. Based on this information and using temperature as the only indicator of biomass production, it was found that the maximum predicted biomass production would be 6.2 g/m²·day and the minimum would be 5.2 g/m²·day. The overall maximum biomass production found was using TOC as an indicator and minimum with temperature as the main indicator, while other studies have shown a maximum productivity of 31 g/m²·day. For the economic analysis, this maximum value was used as a reference to determine if improvements in production could make the system profitable. Therefore, the values used for the calculations were 31 g/m²·day and 5.2 g/m²·day as the maximum and minimum, respectively. This is equivalent to a maximum and minimum value of 30.7 and 16.8 kg of biomass per year for an algal reactor surface area of 9 m², respectively.

Based on the published literature, algae species that naturally grow in agricultural wastewater in New Zealand are summarized in Table 1. In a study performed by Hariz et al. [30], the distribution of naturally growing species in agricultural wastewater in an attached growth system was found to be 35% *Oedogonium* sp., 35% *Spirogyra* sp., 10% *Cladophora* sp., 10% *Melosira* sp., 5% *Oscillatoria* sp., and 5% in a combination of *Klebsormidium* sp, *Ulothrix* sp. and *Stigeoclonium* sp. In theory, one of the biggest indicators of biogas and methane production is the distribution of polymers (lipids, carbohydrates, and proteins). In this study, we used theoretical values [31–34] to predict the biogas production and methane content that could be expected for a combination of species and a best- and worst-case scenario for individual species.

**Table 1.** Algae selection. Biogas and methane yields are based on a worst-case scenario production of 16.8 kg of algal biomass per year and a best-case scenario of 101 kg of biomass per year.

| Species | Protein | Lipid | Carbohydrate | Biogas (m³/year) | | Methane (m³/year) | | Ref. |
|---|---|---|---|---|---|---|---|---|
| | | | | Min | Max | Min | Max | |
| *Oedogonium* sp. | 38% | 13% | 50% | 11.9 | 63.5 | 6.4 | 39.0 | [35] |
| *Spirogyra* sp. | 25% | 9% | 66% | 11.9 | 135.5 | 6.3 | 90.6 | [36] |
| *Cladophora* sp. | 16% | 19% | 65% | 12.6 | 161.2 | 7.0 | 109.6 | [37] |
| *Melosira* sp. | 48% | 40% | 12% | 13.1 | 139.8 | 8.0 | 101.3 | [38] |
| *Oscillatoria* sp. | 62% | 8% | 30% | 11.2 | 80.7 | 5.9 | 55.3 | [39] |
| *Klebsormidium* sp. | 23% | 32% | 29.3% | 11.2 | 140.4 | 6.7 | 99.2 | [40] |
| *Ulothrix* sp. | 13% | 19% | 68% | 12.6 | 165.4 | 7.0 | 112.2 | [37] |
| *Stigeoclonium* sp. | 22% | 18.6% | 43.4% | 10.6 | 125.6 | 5.9 | 86.4 | [41] |

*2.3. Theoretical Bioremediation of Algae*

Another set of equations developed by Fica et al. [11] allow for the prediction of nutrient uptake by algae. These are specific to an RABR and are as follows:

$$K_{N,prediction} = 0.723 \times \theta_N{}^{T_{water}-280} \tag{3}$$

$$K_{P,prediction} = 0.098 \times \theta_P{}^{T_{water}-280} \tag{4}$$

where $K_N$ and $K_P$ represent the nitrogen and phosphorus uptake in mg/m²·day, respectively, where the $\theta_N$ and $\theta_P$ values are 1.0098 and 1.0101, respectively. These were chosen based on the TOC concentration of the effluent as the most similar to 1200 mg/L. The highest and lowest temperatures of 26 and 8 °C were used. The characteristics of dairy effluent in a New Zealand dairy farm are summarized in Table 2 with values from [27].



**Table 2.** Characteristics of liquid dairy effluent from a New Zealand dairy farm after solid separation [27].

| Characteristic | Liquid |
|---|---|
| Volatile solids (VSs) | 1850 g/m$^3$ |
| Total solids (TSs) | 3300 g/m$^3$ |
| Total nitrogen | 290 g/m$^3$ |
| Total ammoniacal-N | 178 g/m$^3$ |
| Nitrate-N + Nitrite-N | <0.10 g/m$^3$ |
| Total Kjeldahl Nitrogen | 290 g/m$^3$ |
| Biochemical oxygen demand [a] | 720 g O$_2$/m$^3$ |
| Chemical oxygen demand | 3000 g O$_2$/m$^3$ |
| Total Carbon | 1380 g/m$^3$ |
| Oil and grease | 310 g/m$^3$ |
| Tannin | 152 g/m$^3$ |
| Total VFA (as acetic acid) | 320 g/m$^3$ |
| Formic acid | <5 g/m$^3$ |
| Acetic acid | 200 g/m$^3$ |
| Propionic acid | 137 g/m$^3$ |
| Butyric acid | 7 g/m$^3$ |
| NDF | 14.7% DM [b] |
| ADF | 9.5% DM [b] |

[a] Carbonaceous biochemical oxygen demand (cBOD5). [b] units are in percentage of DM, DM = dry matter.

### 2.4. Economic Analysis

The rotating algal biofilm reactor consists of two circular ends with a circumference of 6 m and a diameter of 1.9 m. There are eight spokes to every wheel, a total of 16 spokes with one central shaft with a diameter of 159 mm and a length of 2.5 m. There are eight 1.5 m long pieces of aluminum distributed around the cylinder. The cotton rope used in the design is 6.4 mm thick, with a total length of approximately 1400 m.

The total capital cost was estimated based on Formula (5). This formula includes the purchase cost (PC) and direct fixed capital costs (DFCs). The DFC was calculated using Formula (6). This formula includes aspects such as piping, electrical, construction, engineering, contractor's fee, and contingency.

$$\text{Total Capital Cost} = (PC) + (DFC) \tag{5}$$

$$(DFC) = \text{Direct cost (DC)} + \text{Indirect Cost (IC)} + \text{Other Cost (OC)} \tag{6}$$

## 3. Results and Discussion

### 3.1. Biogas Potential According to Different Species

For the anaerobic digestion of the combination of species with the composition found in nature and described previously, a yearly biomass production of 18 kg/unit could result in the best-case scenario production of 17.5 m$^3$ biogas/year and 12 m$^3$ of methane/year. Meanwhile, the worst-case scenario is 9.4 m$^3$ biogas/year and 5.1 m$^3$ methane/year. In terms of individual species and the potential for biogas production, the results found are summarized in Table 1. The best-case scenario in terms of methane production is for *Ulothrix* sp., with the potential production of 112.2 m$^3$ of methane/year. The worst-case scenario is for *Oscillatoria* sp., with a theoretical production of 5.85 m$^3$ of methane per year.

### 3.2. Theoretical Bioremediation of Algae

Equations (3) and (4) were used to calculate nutrient uptake at different temperatures. The total surface area of the RABR is 9 m$^2$ and, assuming that algae can grow all year round due to water temperatures higher than 8 °C, and the number of days where bioremediation occurred was 365. The results are summarized in Table 3.

**Table 3.** Nutrient uptake rate by RABR.

| Component | Units | Min | Max | Avrg | STDV |
|---|---|---|---|---|---|
| Nitrogen | mg/m$^2$ day | 0.731 | 0.871 | 0.801 | 0.098 |
| Phosphorus | mg/m$^2$ day | 0.099 | 0.119 | 0.109 | 0.014 |
| Total nitrogen | g/year | 2.4 | 2.8 | 2.6 | 0.28 |
| Total phosphorus | g/year | 0.32 | 0.38 | 0.35 | 0.04 |

Avrg = average, STDV = standard deviation.

Considering the small amount of nitrogen and phosphorus removed, the system studied did not provide any significant benefits for the bioremediation of dairy effluent. Temperature does not appear to improve the nutrient uptake substantially, considering the standard deviation between the minimum and maximum values is small.

*3.3. Technoeconomic Assessment*

3.3.1. Capital Costs

The maximum and minimum values of biogas production were estimated in the previous section and were used as a starting point for the economic assessment. The total purchase costs include all the equipment that is required to produce biomethane. One of the biggest equipment costs is the motor, followed by the wheel. The rest of the equipment costs are summarized in Table 4.

**Table 4.** TEA of the algae component in the system and equipment cost.

| Component | Price NZD/Unit | QTY | Total Cost (NZD) |
|---|---|---|---|
| Wheel | 1593 | 1 | 1593 |
| Rope | 2112 | 1 | 2112 |
| Floating platform | 2464 | 1 | 2464 |
| Motor | 3840 | 1 | 3840 |
| Harvester | 260 | 1 | 260 |
| Gearbox | 291 | 1 | 291 |
| Bioreactor | 937 | 1 | 937 |
| Total equipment purchase cost (PC) | | | 11,500 |

The direct, indirect, and other costs are calculated by multiplying an established factor in order to estimate the cost for each component. The factors used are as follows: 0.16 for piping, 0.11 for electrical work, 0.34 for construction, 0.32 for engineering, 0.18 for the contractor's fee, and 0.36 for contingency [42]. The total amount for each category is summarized in Table 5.

**Table 5.** Direct fixed capital (DFC) and total capital cost.

| Component | Factor | NZD |
|---|---|---|
| Direct cost (DC) | | |
| Piping | 0.16 × PC | 1840 |
| Electrical facilities | 0.11 × PC | 1265 |
| Total direct cost (DC) | | 3104 |
| Indirect cost (IC) | | |
| Construction | 0.34 × DC | 1056 |
| Engineering | 0.32 × DC | 993 |
| Total indirect cost (IC) | | 2049 |
| Other Cost (OC) | | |
| Contractor's fee | 0.18 × (DC + IC) | 928 |
| Contingency | 0.36 × (DC + IC) | 1855 |
| Total other cost (OC) | | 2783 |
| Direct fixed capital (DFC) | | 7936 |
| Total capital cost | | 19,434 |

The largest cost components are the direct costs (piping and electrical facilities), indirect costs (construction and engineering), and the other costs (contractor's fee and contingency). The direct costs are the costs of materials and labor that are directly used to build the project. In this case, the piping and electrical facilities are the most expensive direct costs. The indirect costs are the costs of overhead and support that are not directly related to the construction of the project. In this case, it was construction and engineering. The other costs are the costs that are not included in the direct or indirect costs. In this case, it was the contractor's fee and contingency. The total capital cost of the project was NZD 19,434.

A summary of the total capital cost distribution is presented in Figure 2. The y-axis shows the different cost components and the x-axis shows the cost in thousands of dollars. The most expensive component is the motor, which costs NZD 3840. The next most expensive components are the floating platform at NZD 2464 and the rope at NZD 2112. These three components make up about 43% of the total project cost.

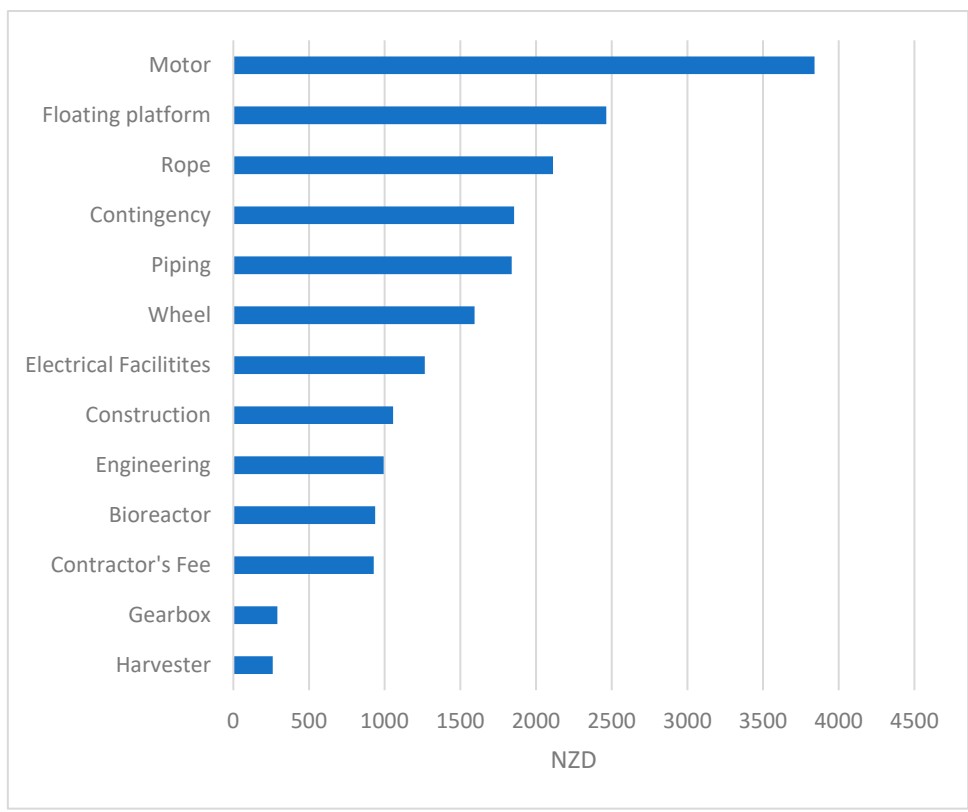

**Figure 2.** Total capital costs.

The other components are all less than NZD 2000 each. These include contingency, piping, wheel, electrical facilities, construction, engineering, bioreactor, contractor's fee, gearbox, and harvester. It is important to note that these are just the capital costs. The project also has operating and maintenance costs, which are not included in the graph but will be discussed in the next section.

### 3.3.2. Operating Costs

The yearly operating costs are summarized in Table 6. Some of the operating costs are calculated as a percentage of the DFC. These are as follows: 1% of the DFC for insurance and 6% of the DFC for maintenance and repairs. The labor is calculated based on 4 h spent harvesting, with the minimum wage being NZD 22.7 hour for 2023 and 18 harvests per year. The number of harvests is based on the optimal harvesting time found by Woolsey et al., 2011 [10], which is 24.5 days. The electricity is calculated based on 639 kwh used/year and

an electrical cost of NZD 0.3/kwh. The operating costs are equal to 10% of the total capital cost. The biggest operating cost is the harvesting labor.

**Table 6.** System operating costs.

| System Operating Costs (NZD/year) | NZD/year |
|---|---|
| Fixed costs | |
| Insurance | 79 |
| Variable costs | |
| Harvesting labor | 1362 |
| Electricity | 192 |
| Maintenance and repair | 476 |
| Total operating costs | 2109 |

There is scope for improvements to the operating costs in terms of harvesting by means of integrating practical knowledge around the material properties of the harvested algae. In particular, this includes the compressibility and permeability parameters described by Wang et al. [43] and Kovalsky et al. [44]. These properties and the characterization methods contained within could form the basis for automating labor-intensive harvesting and should form part of the focus of fundamental studies in the future.

### 3.3.3. Profitability Analysis

A comparison of the best and worst cases of the production of algae was used to measure the different profitability scenarios. As seen in Table 7, neither case is profitable for biomethane generation. To calculate the product value, the cost of natural gas in New Zealand was used. This value was equivalent to NZD 0.15/kwh, and a conversion factor of 0.27 kwh/MJ was used. The most revenue that was possible was NZD 187/year from selling biomethane. Given the low cost of natural gas in New Zealand, the product revenues are not sufficient to justify the expenditure. The DFC of this system is too high in comparison with the potential revenue. This is also the case for the yearly operating costs, and a deficit of NZD 2101 and NZD 1922 was found for the worst and best case. The profitability analysis has shown the system to be uneconomical for its application on a New Zealand dairy farm with biomethane as the main product.

**Table 7.** Profitability analysis of a RABR in a New Zealand farm.

| Component | Min | Max |
|---|---|---|
| Biomass production (kg/year) | 17 | 101 |
| Methane production (m$^3$/year) | 6 | 112 |
| Methane yield (m$^3$CH$_4$/kg biomass) | 0.66 | 1.11 |
| Heating value (MJ/m$^3$) | 35.8 | 39.8 |
| Electrical output (kwh/year) | 58 | 1237 |
| Product (NZD/year) | 9 | 187 |
| DFC (NZD) | 19,434 | 19,434 |
| Operating costs (NZD/year) | 2109 | 2109 |
| Gross profit | −2101 | −1922 |
| Net profit | −1453 | −1275 |
| Return on investment | −7.5% | −6.6% |

In terms of the energy return on investment (EROI), the worst case corresponds to 0.05, and the best case is 1.94. As a point of reference, hydroelectric dams have a value of 40 to 250 [45]. Solar energy has a value of 5.9 and 11.8 approximately [46].

The methane content in biogas produced from algal biomass used for the calculations was 53% for the WC and 68% for the BC. In a study performed by Ansari et al. [47], *Scenedesmus obliquus* was digested with different pretreatments, resulting in methane contents of 45 to 74%. The revenue found in their study was between NZD 0.14 and

0.18/kg of biomass. Meanwhile, in our study, the revenue was NZD 0.5/kg of biomass for WC and NZD 1.9/kg in the BC. In their study, they evaluated the revenue of biogas as the product and compared it to the costs of pretreatment as the total input costs. This analysis resulted in an overall return on investment of NZD 0.086 per kg for dried powdered algae as well as for heat-treated algae. As a point of comparison, using the same approach, with the overall return on investment being the total revenue generated minus the total input cost per year and without the capital cost expenditure being considered, the WC overall return on investment was NZD −111/kg of biomass and the BC was NZD −8/kg of biomass. In our study, some of the biggest yearly costs include labor, which is not included in the total input cost offered by Ansari et al. [47].

Another point of comparison is a study performed by Barlow et al. [48]. Biocrude from algae grown in an RABR on municipal wastewater was studied. Different scenarios were evaluated, from less profitable to an optimized one. The worst-case scenario minimum fuel-selling price was NZD 1.15/MJ, and the best-case scenario was NZD 0.13/MJ [48]. As a point of comparison, the minimum fuel-selling price of methane in this study was NZD 8.47/MJ for WC and NZD 0.24/MJ for BC. The algal production, polymer distribution, and harvesting frequency were the main factors that affected the minimum fuel-selling price of methane.

A study on a 100-cow farm in the United States provides helpful information about different algal systems used on a farm [49]. The scenarios studied were as follows: effluent use for land application, anaerobic digestion, an open pond system (OPS), and an algal turf scrubber (ATS). When evaluated for biogas production, the total capital cost for an anaerobic digestion plant was USD 334,349, with annual operating costs of USD 40,741 and a revenue as bioelectricity of USD 16,827. The systems that were coupled with algal production and the biomass added to the anaerobic digestor with the effluent resulted in an additional capital cost of USD 291,557 for OPS and USD 307,302 USD for ATS. It is important to note that these values do not reflect the cost of land required for algal growth. These values are 3.4 ha of land for an OPS and 2.6 ha of land for an ATS. In New Zealand, the median sale price per hectare of dairy farm is NZD 43,160 [50]. Using algal systems such as OPS and ATS could incur a loss of productive land equivalent to NZD 146,744 and 112,216 NZD, respectively. One of the main advantages of the rotating algal biofilm is that it does not require additional land or water use. In this study, the additional bio-electricity revenue from the digestion of biomass is equivalent to 37% for OPS and 43% for ATS. Meanwhile, the increase in capital cost is 41% for OPS and 44% for ATS. The only way in which the algal systems were profitable was by selling nitrogen credits. These markets are nonexistent in New Zealand, and therefore, both studies, in addition to ours, provide enough evidence to show that algal growth for biogas and/or electricity production on a small dairy farm setting is not profitable.

3.3.4. Economy of Scale

To determine if there were any economic benefits of increasing the capacity of the system, Equation (7) was used as follows:

$$C_2 = C_1 \left( \frac{S_2}{S_1} \right)^n \tag{7}$$

where $C_2$ equals the capital cost of increasing the capacity to 35 units. This is based on the assumption that the pond size for the studied farm is 21 m in width, 30 m in length, and 2 m deep. $S_1$ is equivalent to 1 unit and $S_2$ to 35 units. The $n$-factor used is 0.6. The result was a total capital cost of NZD 164,060 for 35 units instead of NZD 680,191. Even though this represents increased savings, the operating cost for 35 units is at least NZD 27,520, assuming that the harvesting of all units can be supervised by the same person. The total revenue for selling biogas is NZD 6545. The yearly deficit would be at least NZD 20,975. Therefore, the system is not profitable even when the scale of production is increased. The total nitrogen and phosphorus removed by 35 units would be 83 to 99 g N/year and 11 to

13.5 g P/year; therefore, in terms of nutrient uptake, the additional units do not provide additional benefits either.

## 4. Conclusions

The purpose of this study was to evaluate the profitability of an RABR for bioremediation and biomethane production in a New Zealand dairy farm. The proposed system, while tackling some of the limitations of other algal production systems, does not offer an economical solution. A gross profit of NZD −2101 and NZD −1922 for the worst- and best-case scenario confirm this. Some of the biggest costs that are limiting the profitability of the studied system are the motor, labor, the floating platform, and the rope. Further studies on cheaper designs should be conducted. Another option would be to improve economics by growing a species that offers significant added value, such as pharmaceuticals or cosmetics. The main limitation is the bacteria present in the dairy effluent, which needs to be overcome. A significant improvement in the biomass production in RABR would be required to justify the expenditure for biogas production. In this case, using genetically modified species that could be enhanced to produce more biomass, uptake more nutrients, and produce more polymers that enhance biogas production would be required. Different ways of optimizing the cultivation of algae, for example, temperature control could enhance lipid production [51], and different applications could be explored. Nevertheless, an economic analysis should be performed to determine if the cost of controlling the temperature in comparison to profits would be beneficial.

**Author Contributions:** Conceptualization, M.H.-C.; methodology, M.H.-C.; validation, M.L.; formal analysis, M.H.-C.; investigation, M.H.-C.; writing—original draft preparation, M.H.-C.; writing—review and editing, M.H.-C., M.L., P.K. and G.G.; visualization, M.H.-C.; supervision, M.L., P.K. and G.G.; project administration, M.L. and M.H.-C. All authors have read and agreed to the published version of the manuscript.

**Funding:** This research received no external funding.

**Institutional Review Board Statement:** Not applicable.

**Informed Consent Statement:** Not applicable.

**Data Availability Statement:** Raw data is available upon request.

**Conflicts of Interest:** The authors declare no conflicts of interest.

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
