# Peer review of "The Hard Reality of Biogas Production through the Anaerobic Digestion of Algae Grown in Dairy Farm Effluents"

_fermentation, doi:10.3390/fermentation10030137_

Round 1

Reviewer 1 Report

Comments and Suggestions for Authors

The manuscript “The hard reality of biogas production through the anaerobic digestion of algae grown in dairy farm effluent” mainly investigated the biogas production from algae grown in a RABR in the liquid portion of dairy wastewater. The algae culture from waste material is an interesting topic and some suggestions are shown below.

1.      The abstract mainly describes the experimental background and results, and it is recommended to include more important experimental data.

2.      Keywords. Please use the full name of RABR.

3.      Introduction. Please add more examples of using waste materials to cultivate microalgae, compare the differences between this work and previous research, and further emphasize the novelty of this paper.

4.      A recent work on algae culture using waste effluent can be updated, such as “Combined transcriptomic and metabolomic analyses of temperature response of microalgae using waste activated sludge extracts for promising biodiesel production”, to further improve the introduction part.

5.      Line 61. The “sp.” in “Chroococcus sp.” should not be shown in italic form. Please check the whole manuscript.

6.      Materials and Methods. The detailed compositions of dairy farm effluent can be added.

7.      Results and discussion. What are the limitations of the reactor used in this work? How to scale up the application? Please look forward to future development.

Author Response

Thanks for the feedback. Please find in italics responses to your requests. 

  1. The abstract mainly describes the experimental background and results, and it is recommended to include more important experimental data. Please refer to abstract for information request being added.
  2. Keywords. Please use the full name of RABR. Done
  3. Introduction. Please add more examples of using waste materials to cultivate microalgae, compare the differences between this work and previous research, and further emphasize the novelty of this paper. We have added two examples of algae grown in different wastewater and have included the novelty aspect of the article in the last paragraph of the introduction 
  4. A recent work on algae culture using waste effluent can be updated, such as “Combined transcriptomic and metabolomic analyses of temperature response of microalgae using waste activated sludge extracts for promising biodiesel production”, to further improve the introduction part. Have added this article to the conclusion and recommendations for future work.
  5. Line 61. The “sp.” in “Chroococcus sp.” should not be shown in italic form. Please check the whole manuscript. Done
  6. Materials and Methods. The detailed compositions of dairy farm effluent can be added. Done, please refer to the table.
  7. Results and discussion. What are the limitations of the reactor used in this work? How to scale up the application? Please look forward to future development. As mentioned in the results section, the main limitation is the economic aspect for a dairy farmer to implement this in his effluent pond. We included economy of scale but even then the economics are not positive. The only potential changes are in biomass growth and uptake, but natural gas in New Zealand is still very cheap and not worthwhile to produce it from anaerobic digestion of algae with the given economics. Please find more information in the conclusion part of the paper.

Please let us know if you have any additional comments/queries.

Reviewer 2 Report

Comments and Suggestions for Authors

1- The abstract lacks clarity regarding the research methodology and outcomes. It is ambiguous whether this study is based solely on literature data or if it incorporates experimental data generated by the authors. Please revise the abstract and highlight the novel contributions and significance of the current work. Additionally, authors should substantiate their claim regarding the lack of economic viability by providing supporting evidence.

2-Please specify the concept of "bioremediation capacity" in the last three lines of the introduction to ensure clarity for readers.

3-The title "Algae Selection" in subtitle 3.1 appears more appropriate for the materials and methods section. Consider revising the title to better reflect the results obtained from this section.

Author Response

1- The abstract lacks clarity regarding the research methodology and outcomes. It is ambiguous whether this study is based solely on literature data or if it incorporates experimental data generated by the authors.  Please revise the abstract and highlight the novel contributions and significance of the current work. Please find suggestions added to the abstract.

Additionally, authors should substantiate their claim regarding the lack of economic viability by providing supporting evidence. Please refer to line 334 were another study was used as evidence that anaerobic digestion of algae on it's own and whilst also taking into account the costs of land is not profitable on it's own.

2-Please specify the concept of "bioremediation capacity" in the last three lines of the introduction to ensure clarity for readers. Done

3-The title "Algae Selection" in subtitle 3.1 appears more appropriate for the materials and methods section. Consider revising the title to better reflect the results obtained from this section. Done